# Dietary Supplementation with *Pithecellobium dulce* (Roxb) Benth Fruits to Fattening Rabbits

**DOI:** 10.3390/ani13203249

**Published:** 2023-10-18

**Authors:** Jairo Apáez-Barrios, Juan Ocampo-López, Sergio Soto-Simental, Victoria Guadalupe Aguilar-Raymundo, Maricela Ayala-Martínez

**Affiliations:** 1Área Académica de Medicina Veterinaria y Zootecnia, Instituto de Ciencias Agropecuarias, Universidad Autónoma del Estado de Hidalgo, Ex-Hacienda de Aquetzalpa, Ave., Universidad Km 1, Tulancingo de Bravo 43600, Hidalgo, Mexico; jairo.apaez@hotmail.com (J.A.-B.); jocampo@uaeh.edu.mx (J.O.-L.); sotos@uaeh.edu.mx (S.S.-S.); 2Programa Académico de Ingeniería Agroindustrial, Universidad Politécnica de Pénjamo, Carretera Irapuato, La Piedad Km 44, Predio el Derramadero, Pénjamo 36921, Guanajuato, Mexico

**Keywords:** legume, feed, meat quality, carcass trait, sensory analysis, blood biochemistry

## Abstract

**Simple Summary:**

Rabbit meat is considered as a functional food. But, there are few studies directed to determine the influence of natural additives from farm to the table. This study was performed using a fruit obtained from *Pithecellobium dulce* to follow its effect in rabbits from farm to a fresh meat product. The use of this fruit at 5% of the diet increased the dry and organic matter digestibility of the diet and improved feed conversion rate. Also, it increased acceptance of meatballs prepared with rabbit meat obtained from that animal’s feed.

**Abstract:**

*Pithecellobium dulce* produces a fruit used in alternative medicine that could be utilized to feed rabbits. The objective of this study was to measure the effect of the *P. dulce* fruit on productive performance, carcass traits, meat characteristics, and meat product quality as well as shelf-life. Seventy-two California × English pot crossbreed rabbits (35 d age) were randomly distributed into two treatments: a control group without *P. dulce* and another group fed with 5% of *P. dulce,* and fattening for 28 d. Productive performance parameters, blood biochemistry and hematology, apparent digestibility, carcass traits, meat characteristics, and meat product shelf-life were measured. The results indicate inclusion of 5% *P. dulce* improves (*p* < 0.05) dry and organic matter digestibility and feed conversion rate, but some serum blood enzymes were increased (*p* < 0.05). The a* value, hardness, and pH decreased (*p* < 0.05) in the group fed with *P. dulce*. Antioxidant properties in the meatballs were different (*p* < 0.05), improving shelf-life and acceptance in sensory analysis. In conclusion, the use of 0.5% of *P. dulce* fruits to feed fattening rabbits can be used to improve the shelf-life of rabbit meat.

## 1. Introduction

Rabbit production is improving in productivity, which has led to the design of more efficient diets, since rabbits’ feeds are formulated mainly based on by-products high in fiber, animal and vegetable fats, and other ingredients that contain nutrients sufficient for maintaining an efficient productivity [1]. However, rabbit is a species that produces excellent meat, including nutritional characteristics and potential health properties, with this meat and its derivatives considered as functional foods due to their functional compounds [2]. One of the benefits of this species is that it can be fed with different fibrous material, parts of herbs and spices as alternatives to additives or ingredients [3]. In addition, feed costs associated with producing rabbit meat are high which is why alternatives are being sought in order to decrease them. One ingredient which could be an alternative is the fruit from the tree called *Pithecellobium dulce*.

*P. dulce* is a fruit which originates from the Americas in countries including Brazil, Argentina, Colombia, and Mexico, but it is also distributed in several countries around the world, such as India or the tropical regions of Africa. This tree belongs to the Fabaceae family and is one of the 18 species of the genus *Pithecellobium* [4]. Murugesan et al. [5] reviewed therapeutic and biological properties of *P. dulce*, indicating that it has insecticide, anti-diabetic, anti-hyperlipidemic, antioxidant, antiulcer, antidiarrheal, antibacterial, and other properties. Dhanisha et al. [6] demonstrated that an extract of *P. dulce* fruit induced apoptosis in vivo and in vitro. Furthermore, Vargas-Madriz et al. [7] reviewed the antioxidant capacity and phenol profile of *P. dulce* indicating the main phenolics reported are caffeic acid, chlorogenic acid, ferulic acid, gallic acid, p-coumaric acid, protocatechuic acid, apigenin, catechin, daidzein, kaempferol, luteolin, quercetin, myricetin, naringin, and rutin. However, it was also mentioned that antioxidant capacity varies according to all the studies reviewed. *P. dulce* are used in combination with other plants to feed goats, using leaves [8,9] or fruits [10] The above-mentioned findings indicate that the fruit of *P. dulce* is an ingredient that could be used to elaborate animal feed. The objective of this study was to evaluate the effect of *P. dulce* fruit on productive performance, carcass traits, meat characteristics, biochemical and hematology analysis; as well as meatballs prepared with rabbit meat; as a potential alternative ingredient to feed fattening rabbits.

## 2. Materials and Methods

### 2.1. Raw Material and Proximate Analysis

The fruit of *P. dulce* was collected in San Miguel de las Palmas, Guerrero, Mexico. The fruits were dried at 28 °C under shadow, and were then grounded in an Antarix grinder model THCF2800M13 (Antarix de México, Mexico City, Mexico). Afterwards, a proximate analysis was performed according to AOAC methodology [11] to determine moisture (930.15), crude protein (945.01), and crude fat (954.02). Regarding fiber fractions (NDF and ADF), the technique described by Van Soest et al. [12] was used.

### 2.2. Animals and Experimental Design

This study and animal management were carried out according to the institutional committee guidelines on animal care (protocol number CICUA/ICAP 001/2020). The experiments were conducted in a rabbitry located in Tulancingo, Hidalgo, Mexico. Ambiental conditions in the rabbit production house had an average temperature of 17 °C and 70% relative humidity. Seventy-two rabbits were used, which were 35 d of age, unsexed, California × English pot crossbreed, and weighed 650 g on average. The animals were selected and distributed randomly in two treatments, a control group (C, *n* = 36) and a group (G5, *n* = 36) fed with 5% of fruit of *P. dulce*, with nine repetitions (*n* = 4 rabbits). The fattening period was 29 d. In addition, the animals were housed in cages measuring 45 × 40 × 60 cm which were adapted with automatic drinkers and manual feeders. The rabbits were fed ad libitum.

### 2.3. Diets

Diets were prepared following the nutritional requirements of the National Research Council [13], while the ingredient composition was based on the guidelines provided by Fundación Española para el Desarrollo de la Nutricion Animal [14]. Diet formulations had to be isoproteic (16%), isoenergetic (2.5 Mcal·kg^−1^ of digestible energy), and isofibrous (16% Neutro Detergent Fiber) as shown in Table 1. The ingredients were mixed in an ASF model MZ50 double helicoidal mixer (Molinos y Mezcladoras Industriales S. A. de C. V. Mexico), and then pelletized in a SKJ-120 feed pellet machine (Yuezhen Machinery Co., Jinan, China) and finally stored in a hermetic container until use.

### 2.4. Productive Performance

Feed consumption was registered daily (offered and rejected) while live weight was measured every week using a Mettria MTNUV-40 digital scale (Mettria México, Ciudad de México, Mexico). From the data obtained, the average daily feed intake (DFI), daily weight gain (DWG), and feed conversion rate (FCR) were calculated between ages 35 and 63 days. In addition, total weight gain and total feed intake were also determined, including initial and final weight of the experiment.

### 2.5. Apparent Digestibility

Dry matter, organic matter, neutral detergent fiber, and acid detergent fiber were determined according to Perez et al. [15]. Briefly, 8 rabbits by group were selected to perform apparent digestibility, then feces were collected from each cage every morning during the last 4 days of the fattening period. Afterwards, feces were dried in a Riossa model HCF82D oven (RSU Labsupply, Monterrey, NL, Mexico). Content for moisture, ash, neutral detergent fiber (NDF), and acid detergent fiber (ADF) was determined in both feces and feed as indicated above. Subsequently, the digestibility coefficient was calculated.

### 2.6. Carcass Traits

After the fattening period, animals (63 days of age, *n* = 32 rabbits by group) were weighed and transported to the meat laboratory belonging to the Instituto de Ciencias Agropecuarias and then slaughtered without previous fasting and mechanical concussion stunning according to national legislation [16]. Before slaughtering, the dorsal length and the lumbar circumference of the animals was measured (from the atlas to the last ischia vertebra) using a measuring tape. After evisceration, the weights of the skin, feet, hot carcass, viscera (including esophagus, trachea, digestive apparatus, heart, lungs, kidneys, and liver), carcass length, and lumbar circumference were obtained. Carcasses were stored in refrigeration at a temperature of 4 °C for 24 h. Afterwards, cold carcass and main cuts (head, forequarter, thoracic cage, foreleg, and legs) were obtained according to the indications provided by Blasco et al. [17]. Then, legs were dissected into meat, fat, and bone using a Scout Pro model SP402 scale (Ohaus Corporation, Pine Brook, NJ, USA).

### 2.7. Hematological and Biochemical Analysis

During the exsanguination procedure, 3 mL of blood were collected in vacutainer tubes (*n* = 9 by treatment) to determine blood biochemical analysis using BA400 Biosystem equipment. Another tube was used to collect blood and perform blood biometry using a hematology analyzer Procyte Dx (Idexx laboratories Inc., Westbrook, ME, USA).

### 2.8. Meat Characteristics

The carcasses were kept for 24 h under refrigerated conditions, then color was measured using a Minolta colorimeter model CM-580d (Konica-Minolta, Tokyo, Japan) with a CIEL*a*b* color space using an illuminant D65, and 0.8 cm aperture size. The observer was set to 10° according to the American Meat Science Association meat color measurement guidelines [18]. For measuring pH, a Hanna HI99163 meat pHmeter (Hanna Instruments, Cluj-Napoca, Romania) was used. Furthermore, in order to determine water holding capacity (WHC), a technique described by Honikel [19] was employed. Cooking loss was measured by cutting half of a loin which was then weighed and cooked in a hot water bath at 80 °C for 20 min. Subsequently, 1 cm^3^ meat cubes were analyzed for a texture profile analysis in a Brookfield CT3 texture analyzer (Brookfield, Middleboro, MA, USA). The equipment was adapted with a TA3/1000 probe and set up to compress the sample at 50% using a crosshead speed of 1 mm·s^−1^. The sample was compressed twice. Force–time graph parameters of hardness, resilience, cohesiveness, springiness, and chewiness were obtained using Texture Pro CT software (Brookfield, Middleboro, MA, USA).

### 2.9. Analysis of Meatball

Meat obtained from legs was ground in a Torrey grinder (Torrey, Monterrey, NL, Mexico); the meat was separated into two batches. The meatballs were prepared by adding 10 g of salt and 200 mL of water to 1 kg of rabbit meat, which were then mixed. Afterwards, 50 g portions were made and stored on plastic trays, covered with film, and then stored at 4 °C until analysis.

To determine the effect of *P. dulce*, microbiological and physicochemical analysis was performed on days 0, 7, and 14, with dilutions and bacterial counts analyzed using indications according to national legislation [20]. Total viable counts of bacteria *Enterobacteriaceae* and *Staphylococcus* were tested. Antioxidant activity was determined according to Brand-Williams et al. [21] with 2,2-Diphenyil-1-picrylhydrazyl (DPPH) as radical, while antioxidant activity was expressed in mg·mL^−1^. The pH was measured using a Hanna meat pHmeter model HI99163 (Hanna Instruments, Cluj-Napoca, Romania). Finally, water activity (Aw) was determined with a HP23-aw HygroPalm (Rotronic Measurement Solutions, Bassersdorf, Switzerland).

Meatballs were subjected to a sensory analysis using an affective hedonic test to determine the acceptability of the meat’s taste. A total of 120 consumers with an average of 21.5 years participated, of which 41.7% were female and 57.5% were male. A hedonic 7-point scale affective test (7 like very much and 1 dislike very much) was undertaken to determine acceptability. The test was developed according to the indications provided by Drake [22].

### 2.10. Statistical Analysis

In this work, a completely randomized design was used to analyze productive performance parameters, including total weight gain, total feed consumption, and feed conversion ratio; for these variables, treatment was the fixed effect and cage was random term. On apparent digestibility of dry matter, organic matter, neutral detergent fiber, and acid detergent fiber, all carcass traits, all meat characteristics, all biochemical and hematology analysis, an analysis of variance following the general linear model procedure was carried out, continuing with a lsmeans option, using treatment as fixed effect and cage as random term. Statistical model was following:Y_ij_ = μ + β_i_ + ε_ij_ + δ_ijk_,
where Y_ij_ = dependent variable, μ = mean of the variable, β_i_ = the fixed effect of i-th rabbit of the group, δ_k_ = the random effect of k-th cage, and ε_ij_ = experimental error associated with the observation Y_ij_.

A repeated time one-way design was used to analyze feed consumption and daily weight gain through the fattening period; treatment and time was used as fixed effect, with the nested time in treatment, then cage was utilized as random term. Also, all variables measured to determine meatballs’ shelf-life were analyzed through a repeated time one-way design, where treatment and time was used as fixed effect, with the nested time in treatment, then batch was utilized as random term.
Yijk = μ + β_i_ + τ_j_+ β_i(τj)_ + ε_ijk_ + δ_ijkl_,
where Y_ij_ = dependent variable, μ = mean of the variable, β_i_ = the fixed effect of i-th rabbit of the group, τ_j_ = time and β_i_(τ_j_) time inside treatment, δ_l_ = the random effect of l-th cage or batch, and ε_ijk_ = experimental error associated with the observation Y_ijk_.

For sensory analysis, firstly a statistical descriptive analysis was carried out, subsequently a Student’s *t*-test was conducted to determine differences between samples of meatballs elaborated with meat from rabbits given feed with or without *P. dulce*, one session in approximately two hours was conducted, then data collected as indicates above.

The statistical models were the following:$$t=\frac{\left({\bar{x}}_{1}-{\bar{x}}_{2}\right)}{\sqrt{s^{2}\left(1/n_{1}+1/n_{2}\right)}}$$
where *t* is *t*-test, the numerator is the difference between the two consumers groups to taste meat, denominator is an estimate of the standard error. All analyses were performed using SAS software ver. 9.0.

## 3. Results

### 3.1. Proximate Analysis

A proximate analysis of *P. dulce* fruits revealed 89.5% of dry matter, 6.4% ash, 6.2% ethereal extract, 21.5% protein, 74% of NDF, and 32.1% acid detergent fiber. According to the results, *P. dulce* fruits are a rich source of protein and fiber.

### 3.2. Productive Performance

Growth performance parameters are shown in Table 2. Supplementing the rabbits’ diets with *P. dulce* fruits did not affect productive performance (*p* > 0.05), including body weight, feed intake, and feed conversion rate. These results indicate that *P. dulce* fruits could be used to feed growing rabbits.

### 3.3. In Vivo Apparent Digestibility

The results of digestibility are shown in Table 3. Dry matter and organic apparent digestibility were higher (*p* < 0.05) when rabbits consumed 5% of *P. dulce* fruits. However, NDF and ADF digestibility were similar (*p* > 0.05).

### 3.4. Biochemical and Blood Analysis

Results are shown in Table 4. Rabbits of both experimental groups showed similar (*p* > 0.05) compositions from biochemical and blood analyses, except for alanine transferase, aspartate transferase, and phosphorus which were higher (*p* < 0.05) in the G5 group.

### 3.5. Carcass Traits

Carcass traits are shown in Table 5, nearly all of which were similar (*p* > 0.05) between treatments. Nevertheless, feet weight was lower (*p* < 0.05) in G5 group, while drip loss was higher (*p* < 0.05) in the control group.

### 3.6. Meat Characteristics

All the carcass characteristics of the rabbits fed with 5% of *P. dulce* fruits were similar (*p* > 0.05) to the control group (Table 6), except for a* value (redness) and hardness of meat, as lower values (*p* < 0.05) for these parameters were presented in the group fed with *P. dulce*.

### 3.7. Meatball Quality

Measurements of meatballs elaborated with rabbit meat from animals fed with *P. dulce* are shown in Table 7. An interaction treatment with storage time was observed in pH, Aw, and antioxidant activity. However, bacteria count groups were different (*p* < 0.05) over storage time. The pH value increased during storage time, but it was lower in treatment G5 on day 14. The water activity of meatballs in the control treatment decreased during storage time, but G5 treatments were similar during 14 d of storage. Antioxidant activity increased during storage time; however, G5 treatment was higher after 14 d of storage.

### 3.8. Sensory Analysis of Meatballs

The results from the affective hedonic sensory test on meatballs with meat from rabbits fed with P. dulce are shown at Figure 1, with panelists considering that the samples were similar (*p* > 0.05). Therefore, *P. dulce* fruits can be incorporated into rabbit feed because they did not show any adverse effects on sensory analysis.

## 4. Discussion

There is little information about *P. dulce* fruits’ proximate composition. However, there are some studies that revealed chemical composition is diverse; when the fruit is divided into its main parts (seed, aril, and husk), the seed has 39% protein, while dry aril is between 12 and 15%, and fresh aril is 3%; which indicates that composition is influenced by the part of the fruit analyzed [23,24]. Yet, whole *P. dulce* fruits have a high quantity of protein (21.5%) and NDF (78%); according to these amounts, those fruits can be used to feed animals as an additive or ingredient.

Productive performance results indicate the use of *P. dulce* fruits to feed rabbits saw similar results between groups. However, there are several studies on the use of fruits, vegetables, extracts, or essential oils to feed fattening rabbits. Some of these investigations concluded that there are no effects on productive performance, such as the studies by Perna et al. [25] which used cauliflower leaves, Mancini et al. [26] who fed rabbits with quebracho and chestnuts tannins mixes, as well as Kovitvadhi et al. [27] who focused on *Lythrum salicaria*. Other studies indicated an increase in some productive performance parameters, such as Khalid et al. [28] who observed that Moringa oleifera leaf powder increased daily gain and improved the feed rate conversion. In another study, Elwan et al. [29] fed rabbits with 1 and 2% of Citrus limon powder and found that the parameters related to final weight, weight gain, and average daily gain all increased. Perhaps, *P. dulce* fruits did not improve productive performance, but they can be used to feed rabbits.

Apparent digestibility coefficients were similar to other ingredients in rabbit feeds, including bilberry pomace [30]. It is notable that cellular content has high digestion influenced by *P. dulce* fruits. It is possible that this fruit has small quantities of tannins (according to Roselin and Parameshwari [31]); their study also reviewed bioactive compounds in *P. dulce* which indicated the presence of tannins. Also, according to Mancini et al. [26], tannins did not influence the palatability of the feed. However, in contrast to this study, several studies indicate that dry or organic matter digestibility is not influenced by tannins. For example, Kovitvadhi et al. [27] stated that low levels of tannins did not affect digestibility, while similar results were obtained by Dabbou et al. [30]. Dalle-Zotte et al. [32] did not find any differences in dry matter and organic matter digestibility when feeding rabbits with chestnut hydrolysable tannins.

In general, the reference values for these biochemical parameters were normal with regard to rabbits; however, control groups had low levels for alanine transferase, aspartate transferase, and phosphorus compared to G5 group. It is possible that a hepatic malfunction was a consequence of intensive production or use of *P. dulce*. However, biochemical parameters were normal as indicated by Brandão et al. [33]. Aljohani and Abduljawad [34] reported similar concentrations of alanine and aspartate transferase in rabbits fed with *Moringa oleifera*, although values reported in their study were higher than those obtained in this experiment. In contrast, Imbabi et al. [35] found aspartate transferase decreased in rabbits fed with fennel oil, although alanine transferase was similar.

Most of the carcass traits were similar between groups, except feet and drip loss (the growth of animals is related to productive performance; if these last parameters are analogous between groups, it is possible that organs and body composition were similar.) There are other studies that report similar results, such as Wolf and Cappai [36] who used rabbit feed incorporated with acorns (*Quercus pubescens* Willd.) and there was no difference in carcass traits. However, the use of Moringa olefira leaves increased carcass traits in a study by Aljohani and Abduljawad [34]. Moreover, drip loss is related to protein and the accumulation of lactic acid [37], meaning that it is possible that *P. dulce* slightly increased protein content while the pH also remained slightly high.

Meat characteristics of rabbits fed with *P. dulce* fruits were similar between groups, except a* value and hardness. The maximum shear force is correlated with the connective tissue [38]. It is possible that *P. dulce* induced a lower proportion of collagen instead of structural proteins; Sembiring et al. [39] demonstrated that *Muntingia calabura* extract leaves increase collagen density. However, other studies using leaves or fruits did not find differences, such as Pałka et al. [40] who supplemented rabbit feed with *Urtica dioica* L. or *Trigonella foenum-graecum* L., or Garcia-Vazquez et al. [41] who used an infusion of *Chenopodium ambroisoides.* Also, the change in redness color value is associated to Fe and anthocyanins content in the aryl of *P. dulce* fruits, as indicated by Rao et al. [23].

*P. dulce* fruits feed to rabbits decreases Staphylococcus counts. Koné et al. [42] observed that plant extracts or essential oils supplemented into rabbits’ diets decreased bacterial content in rabbit meat under anaerobic conditions. Mancini et al. [43] used 3.5% of turmeric powder to extend the shelf-life of rabbit burgers.

The acceptance of rabbit meat burgers was similar between treatments. Other studies have observed similar results using bilberry pomace [44] and goji berries [45], while Mancini et al. [46] used *Zingiber officinale roscoe* powder in rabbit meat burgers to improve sensorial characteristics.

## 5. Conclusions

The inclusion of *P. dulce* at a proportion of 5% into the diet for growing rabbits did not affect productive performance, carcass traits, or meat characteristics. Regarding biochemical analysis and hematological analyses, there were no differences in almost all parameters evaluated; however, some enzyme levels (alanine and aspartate transferase) were higher in G5. As the product elaborated from rabbit meat with animals fed *P. dulce* after 14 d of refrigeration storage showed a low pH value, a slightly low bacteriological group, and a higher antioxidant capacity, then these parameters are indicatives of the increased shelf-life of the meat product, while sensory analysis showed no differences between the control and treatments. Therefore, this study suggests that the incorporation of *P. dulce* could increase the shelf-life of meat without any deleterious effect on acceptance of the meat by panelists.

## Figures and Tables

**Figure 1 animals-13-03249-f001:**
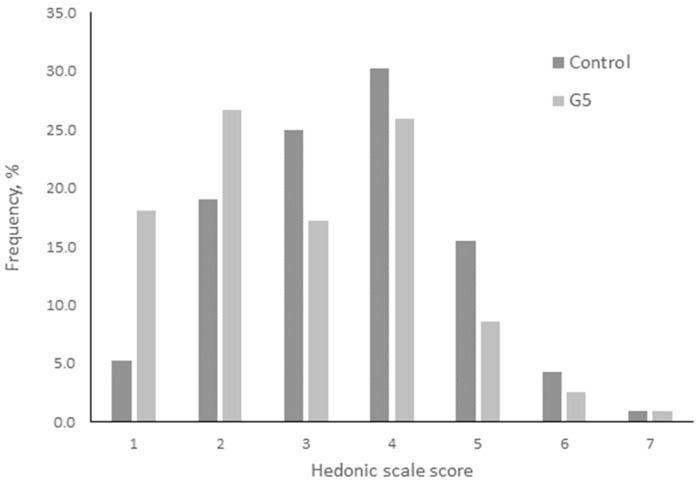
Frequency of acceptability of meatballs elaborated with meat obtained from rabbits fed with *Pithecellobium dulce* fruits.

**Table 1 animals-13-03249-t001:** Diet formulation and chemical composition.

Ingredients (g·kg^−1^)	Treatments ^1^	
	C	G5
Barley straw	13.5	11.0
Corn ground	21.5	22.0
Sorghum ground	12.4	12.4
Dry Distilled Grains	6.5	6.0
Canola meal	4.0	3.0
Wheat bran	8.1	8.1
Canola oil	1.0	1.0
Molasses	2.5	2.5
Soybean meal	15.0	14.0
Soybean hulls	12.5	12.0
Minerals and vitamins premix	3.0	3.0
*Pithecellobium dulce*	0	5.0
Chemical composition, %		
Dry matter	89.5	88.0
Ash	5.7	5.6
Ether extract	6.2	5.4
Protein	15.7	15.7
Acid detergent fiber	45	43
Neutral detergent fiber	60.1	58.3
Digestible energy, Mcal kg MS^−1^	2.6	2.6

^1^ C = control, G5 = *Pithecellobium dulce* fruits at 5% of the diet.

**Table 2 animals-13-03249-t002:** Productive performance of rabbits that consumed *Pithecellobium dulce* fruits.

Productive Parameter	Treatments ^1^		SEM ^2^	*p* Value
	C (*n* = 32)	G5 (*n* = 32)		
Initial weight at 35 d, g	598	576	49	0.99
Final weight at 63 d, g	1596	1583	75	0.42
Average daily weight gain, g	35.64	35.96	2.88	0.28
Average daily feed intake, g	83.25	81.82	5.82	0.26
Feed conversion ratio	2.89 ^a^	2.49 ^b^	0.32	0.19
Total weight gain, g	998	1007	67	0.28
Total feed intake, g	2331	2291	109	0.80

^ab^ Different superscripts between columns indicate significant differences (*p* < 0.05). ^1^ C = control, G5 = *Pithecellobium dulce* fruits at 5% of the diet. ^2^ Standard error of the mean.

**Table 3 animals-13-03249-t003:** In vivo total tract apparent digestibility of diets including *Pithecellobium dulce* fruits consumed by rabbits.

Digestibility Coefficient	Treatments ^1^		SEM ^2^	*p* Value
	C (*n* = 8)	G5 (*n* = 8)		
Dry matter	0.63 ^b^	0.65 ^a^	0.09	0.05
Organic matter	0.67 ^b^	0.69 ^a^	0.08	0.03
Neutral detergent fiber	0.37	0.36	0.09	0.20
Acid detergent fiber	0.59	0.59	0.09	0.79

^ab^ Different superscripts between columns indicate significant differences (*p* < 0.05). ^1^ C = control, G5 = *Pithecellobium dulce* fruits at 5% of the diet. ^2^ SEM = Standard error of the mean.

**Table 4 animals-13-03249-t004:** Blood biochemistry and hematology of rabbits fed with *Pithecellobium dulce* fruits.

Carcass Trait	Treatments ^1^		SEM ^2^	*p* Value
	C (*n* = 9)	G5 (*n* = 9)		
**Blood analysis**				
Hematocrit, L·L^−1^	0.42	0.42	0.01	0.97
Hemoglobin, g·L^−1^	129.57	131.11	3.19	0.73
Red blood cells, ×10^12^	5.68	5.78	0.19	0.72
Mean corpuscular volume, FI	71.42	70.42	68.54	0.53
Mean corpuscular hemoglobin concentration, g·L^−1^	318.57	319.78	2.45	0.73
Platelets, ×10^9^·L^−1^	283.85	365.89	56.29	0.32
Total proteins, g·L^−1^	54.71	55.22	1.43	0.80
White blood cells, ×10^9^·L^−1^	5.20	6.78	1.02	0.29
Neutrophiles, ×10^9^·L^−1^	2.84	4.17	0.75	0.23
Lymphocytes, ×10^9^·L^−1^	2.17	2.43	0.28	0.53
Monocytes, ×10^9^·L^−1^	0.14	0.17	0.04	0.54
**Biochemical analysis**				
Glucose, mmol·L^−1^	7.69	7.90	0.30	0.63
Creatinine, µmol·L^−1^	83.90	86.31	4.15	0.68
Cholesterol, mmol·L^−1^	1.69	2.11	0.15	0.07
Bilirubin, µmol·L^−1^	4.64	8.32	1.69	0.14
Alanine transferase, UI·L^−1^	32.57 ^b^	49.77 ^a^	4.86	0.02
Aspartate transferase, UI·L^−1^	38.14 ^b^	48.55 ^a^	3.30	0.04
Alkaline phosphatase, UI·L^−1^	216.85	211.77	20.20	0.86
Total protein, gL·L^−1^	52.57	51.88	1.80	0.79
Albumin, gL·L^−1^	30.71	30.22	3.02	0.91
Globulin, gL·L^−1^	21.85	18.44	1.38	0.10
Calcium, mmol·L^−1^	3.19	3.16	0.09	0.83
Phosphorus, mmol·L^−1^	2.52 ^b^	2.85 ^a^	0.11	0.04
Lactate dehydrogenase, UI·L^−1^	702.00	903.11	121.95	0.26

^1^ C = control, G5 = *Pithecellobium dulce* fruits at 5% of the diet. ^2^ SEM = Standard error of the mean. ^ab^ Different literals between columns indicate significant differences (*p* < 0.05).

**Table 5 animals-13-03249-t005:** Carcass traits of the rabbits fed with *Pithecellobium dulce*.

Carcass Trait	Treatments ^1^		SEM ^2^	*p* Value
	C (*n* = 32)	G5 (*n* = 32)		
Live weight, g	1562.19	1566.05	80.55	0.97
Dressing out percentage, %	51.30	50.26	0.94	0.44
Body length, cm	30.00	28.55	0.73	0.16
Body lumbar circumference, cm	20.54	20.16	0.65	0.67
Skin, g·kg^−1^ LW ^3^	148.00	146.86	2.61	0.75
Feet, g·kg^−1^ LW	27.90 ^a^	25.01 ^b^	0.81	0.01
Carcass length, cm	28.94	28.03	0.68	0.34
Carcass lumbar circumference, cm	14.75	14.89	0.52	0.84
Viscera ^4^, g·kg^−1^ LW	252.98	259.47	8.58	0.59
Hot carcass weight, g	809.69	790.05	47.24	0.77
Cold carcass weight, g	790.94	759.21	46.59	0.63
Drip loss, %	2.35b	3.47 ^a^	0.31	0.01
Kidney fat, g·kg^−1^ HCW ^5^	11.36	12.05	1.80	0.78
Scapular fat, g·kg^−1^ HCW	3.93	3.35	0.54	0.45
Head, g·kg^−1^ HCW	122.33	114.18	8.84	0.51
Forepart weight, g·kg^−1^ HCW	254.12	232.97	20.09	0.46
Intermedia part weight, g·kg^−1^ HCW	107.28	95.47	9.07	0.36
Hind part weight, g·kg^−1^ HCW	182.08	163.12	17.52	0.45
Legs, g·kg^−1^ HCW	363.35	330.77	29.36	0.43
Meat, g·kg^−1^ Legs	561.66	550.69	36.80	0.83
Bone, g·kg^−1^ Legs	234.12	225.11	10.99	0.56
Dissectible fat, g·kg^−1^ Legs	4.28	5.23	1.01	0.51

^ab^ Different superscripts between columns indicate significant differences. ^1^ C = control, G0.5 = *Pithecellobium dulce* fruits at 5% of the diet. ^2^ SEM = Standard error of the mean (*p* < 0.05). ^3^ LW = live weight. ^4^ Viscera includes esophagus, trachea, digestive apparatus, heart, lungs, kidneys, and liver. ^5^ HCW = Hot carcass weight.

**Table 6 animals-13-03249-t006:** Meat characteristics of the rabbits fed with *Pithecellobium dulce* fruits.

Carcass Trait	Treatments ^1^		SEM ^2^	*p* Value
	C (*n* = 32)	G5 (*n* = 32)		
L*	47.28	47.63	0.41	0.54
a*	1.27 ^a^	0.28 ^b^	0.14	0.00
b*	7.52	7.29	0.18	0.37
C*	7.72	7.42	0.18	0.25
h*	1.51	1.52	0.01	0.44
Water holding capacity	19.91	19.26	0.87	0.60
pH	5.96 ^a^	5.97 ^b^	0.02	0.79
Cooking loss, %	8.28	8.86	0.74	0.58
Hardness, N	24.66 ^a^	19.31 ^b^	0.92	0.00
Adhesiveness, N (×10)	0.43	0.31	0.05	0.08
Resilience	0.17	0.18	0.01	0.16
Cohesiveness	0.49	0.49	0.01	0.74
Springiness	0.57	0.56	0.01	0.53
Chewiness, N	7.28	6.61	0.40	0.24

^1^ C = control, G5 = *Pithecellobium dulce* fruits at 5% of the diet. ^2^ SEM = Standard error of the mean. ^ab^ Different literals between columns indicate significant differences (*p* < 0.05).

**Table 7 animals-13-03249-t007:** pH, aw, microbial analysis, and antioxidant activity of meatball quality elaborated from rabbits that consumed *Pithecellobium dulce* fruits during storage time.

Parameter	Treatment ^1^	Time (d)			SEM ^2^	*p* Value		
		0	7	14		*T*	*t*	*T* × *t*
pH	C	5.823 ^b^	5.890 ^b^	6.703 ^aA^	0.020	0.002	0.001	0.001
	G5	5.907	5.830	6.203 ^B^	0.020			
Aw ^3^	C	0.980 ^a^	0.957 ^b^	0.967 ^a^	0.004	0.210	0.101	0.024
	G5	0.963	0.950	0.967	0.004			
TCV ^4^, Log UFC·g^−1^	C	3.085 ^c^	6.756 ^b^	9.619 ^a^	0.093	0.938	0.001	0.270
	G5	3.213 ^c^	6.660 ^b^	9.553 ^a^	0.152			
*Staphylococcus*, Log UFC·g^−1^	C	2.971 ^c^	7.650 ^b^	10.826 ^a^	0.077	0.001	0.001	0.705
	G5	2.553 ^c^	6.460 ^b^	10.320 ^a^	0.125			
*Enterobacteriaceae*, Log UFC·g^−1^	C	2.649 ^c^	7.835 ^b^	12.661 ^a^	0.060	0.615	0.001	0.521
	G5	2.400 ^c^	8.043 ^b^	12.593 ^a^	0.148			
Antioxidant activity (mg·mL^−1^)	C	11.301 ^a^	4.469 ^bB^	3.825 ^bB^	0.463	0.344	0.001	0.001
	G5	10.066 ^a^	5.908 ^bA^	5.229 ^bA^	0.463			

^ab^ Different superscripts between columns indicate significant differences (*p* < 0.05). ^AB^ Different superscripts between raw at the same time indicate significant differences (*p* < 0.05). ^1^ C = control, G5 = *Pithecellobium dulce* fruits at 5% of the diet. ^2^ SEM = Standard error of the mean. ^3^ Aw = Water activity. ^4^ TVC = Total counts viable bacteria.

## Data Availability

All original data can be obtained or requested from Maricel Ayala Martinez.

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
