# Peer review of "Dietary Supplementation with Pithecellobium dulce (Roxb) Benth Fruits to Fattening Rabbits"

_animals, 2023, doi:10.3390/ani13203249_

Round 1

Reviewer 1 Report

Jairo et al. investigated the effect of dietary supplementation of Pithecellobium dulce fruit products on fatting rabbits in relation to the production traits, blood measures, meat characteristics, and others. Overall, this study was properly designed and performed, and the results could guide us to explore novel dietary supplements in meat rabbits. However, some revisions need to be addressed before accepting for publication. 

L20: The first sentence needs to be rephrased. I think it is not a widely acknowledged conclusion. 

L22-23: How old are these rabbits when selected? How long is the experimental period? The related important information should be indicated here. 

Introduction: Beside rabbit, are there other livestock species in literature subjected to dietary supplementation of Pithecellobium dulce? Overall, the introduction section had not been prepared sufficiently, such as how to support the first sentence in Abstract, and what to make the 5% supplementation used in this study. 

L77-79: How many rabbits did you have at the end of feeding experiment for each group? 

L124: How many samples were collected for hematological and biochemical analysis? 

L144-160: I suggest authors to combine the 2.9, 2.10, and 2.11 sections together. 

L170: The related abbreviations of traits need to be kept constant throughout manuscript. 

L168-185: It is hard to understand how these statistical analyses were conducted for different traits. I suggest to reorganize the related description. 

L190 and L195: Which fixed effects were used? 

In Table 2: In addition to the mean, the sample size measured, standard deviation of traits, and P value of inter-group comparisons need to be given for each trait. This is same for other tables and Figure 1. Furthermore, the content listed here is different from that in Abstract, such as inconsistent conclusion about the feed conversion rate. 

L211-212: How did you conclude that “These results indicate that P. dulce fruits could be used to feed growing rabbits.” as no significant effect was observed regarding the productive performance.

Author Response

I enclose a file.

Reviewer 2 Report

In my opinion, manuscript entitled ,,Dietary supplementation with Pithecellobium dulce (Roxb) Benth fruits to fattening rabbits” can be cosidered, after a few minor corrections, to be published in Animals journal. Below are my comments:

- lines 27-28: did you mean: increased levels of serum blood enzymes?

- In Europe, where I live, rabbits are fattened for more than 29 days. It depends on the breed/line but anyway rabbits should weigh about 2.5 kg. Why did you fatten your rabbits for such a short time? Is it economically justified?

- line 125: I feel like this sentence is incomplete.

- line 182: improved?

- line 214: I think you should use italic letters for latin name.

- Table 2: I think you should recalculate average daily weight gain.

- lines 225-226: The sentence ,,Biochemical and hematological tests can be an important tool…” is not needed in this section (if at all).

I congratulate the authors on a good research idea!

Author Response

I enclose a file

Reviewer 3 Report

GENERAL COMMENT:

I consider this work is within the scope of “Animals”. It contains information useful in a field in which available information on the use of Pithecellobium dulce is scarce and of special interest for searching alternatives to improve rabbit meat quality. Overall, it is well organised, but improvement is needed because there are several flaws. I indicate below points to be improved in the manuscript.

TITLE:

Type “Pithecellobium dulce” in italics.

SIMPLE SUMMARY:

It is OK.

ABSTRACT:

Lines 21, 24, 26, 28, and 30: Type “P. dulce” in italics.

It is OK.

KEYWORDS:

I suggest adding a keyword: “blood biochemistry”.

INTRODUCTION:

Overall, this section is OK. However, some improvement is needed.

Line 35, 36: You say that: “Rabbit production, as with other species involved in meat production, competes with the human population for food from cereals and legumes”.  However, this is only partly true, because great part of rabbit feeding is based on high fibrous raw material not competing with humans. Balanced feed for rabbit usually are formulated with animal and vegetable fats, molasses, beet, apple and citrus pulp, as well as soy hulls, cereal co-products, Lucerne hay, lignified fibrous co-products, and cereal grains (at 100-200 g/kg) and protein concentrates (at 120-220 g/k). Therefore, the level of competence y lower than in other monogastric species. I recommend indicating it and “smoothing” the affirmation that rabbit feeding competes with human food “as with other species”, because rabbits compete to a lesser extent.

MATERIALS AND METHODS:

This section has several flaws. Several improvements are needed.

Lines 81-82: “the animals were housed in cages measuring 80 45x40x60”. However, it is unclear whether the animals were housed individually or not. If not, please indicate the number of rabbits per cage.

Line 82: Type “ad libitum” in italics.

Line 125: The sentences is cut, incomplete. The description of hematological and biochemical analysis is incomplete.

Lines 146-147: For meatballs preparation, it is indicated that “The meatballs were prepared by adding 10 g of salt and 200 mL of water which 146 were then mixed”, but it is necessary to indicate to what amount of rabbit meat were these ingredients added.

RESULTS SECTION:

Overall, this section is OK.

DISCUSSION SECTION:

Overall, this section is OK.

CONCLUSIONS:

The conclusions must be rewritten, because it is indicated that “The inclusion of P. dulce at a proportion of 5% into the diet for growing rabbits did not affect productive performance, carcass traits, or meat characteristics. Regarding biochemical analysis and hematological analyses, there were no differences in the parameters evaluated”, while, according with your results, the inclusion of 5% P. dulce improve (P < 0.05) dry and organic matter digestibility, feed conversion rate, and serum blood enzymes. The a* value, texture profile analysis, and pH decrease (P < 0.05) in group fed with P. dulce…

REFERENCES SECTION:

In general terms, this section need a better adjusted to the style and format of the journal for references. I recommend reviewing it for removing typos and correct several flaws. For example:

Latin names of the organisms must be typed in italics. For example, Line 384: Pithecellobium dulce, and more.

Journal names must be abbreviated.

Line 477: The reference is incomplete.

TABLES:

Table 2: Average daily gain values musty be provided with not more than two decimal digits.

FIGURE: It is OK.

Author Response

I enclose file

Round 2

Reviewer 1 Report

I have no additional comments!